# ^1^H LF-NMR Self-Diffusion Measurements for Rapid Monitoring of an Edible Oil’s Food Quality with Respect to Its Oxidation Status

**DOI:** 10.3390/molecules27186064

**Published:** 2022-09-16

**Authors:** Tatiana Osheter, Salvatore Campisi-Pinto, Maysa T. Resende, Charles Linder, Zeev Wiesman

**Affiliations:** Phyto-Lipid Biotechnology Lab (PLBL), Department of Biotechnology Engineering, Faculty of Engineering Sciences, Ben Gurion University of the Negev, Beer-Sheva 84105, Israel

**Keywords:** edible oils, oxidation, TD-NMR, H relaxation, self-diffusion

## Abstract

The food quality of edible oils is dependent on basic chemical and structural changes that can occur by oxidation during preparation and storage. A rapid and efficient analytical method of the different steps of oil oxidation is described using a time-domain nuclear magnetic resonance (TD-NMR) sensor for measuring signals related to the chemical and physical properties of the oil. The degree of thermal oxidation of edible oils at 80 °C was measured by the conventional methodologies of peroxide and aldehyde analysis. Intact non-modified samples of the same oils were more rapidly analyzed for oxidation using a TD-NMR sensor for 2D T_1_-T_2_ and self-diffusion (D) measurements. A good linear correlation between the D values and the conventional chemical analysis was achieved, with the highest correlation of R^2^ = 0.8536 for the D vs. the aldehyde concentrations during the thermal oxidation of poly-unsaturated linseed oils, the oil most susceptible to oxidation. A good correlation between the D and aldehyde levels was also achieved for all the other oils. The possibility to simplify and minimize the time of oxidative analysis using the TD NMR sensors D values is discussed as an indicator of the oil’s oxidation quality, as a rapid and accurate methodology for the oil industry.

## 1. Introduction

This present study focuses on the rapid and non-destructive monitoring of an edible oil’s quality and safety, with emphasis on the oxidation status of the oils exposed to thermal oxidation [1,2]. It is increasingly recognized that monitoring oxidation in numerous food products at different stages of harvesting, preparation, shipping, storage, and cooking is of importance for achieving a food product’s optimal health and nutritional value [3]. In these applications, low-field (LF) ^1^H NMR energy relaxation measurements are becoming a relatively low-cost facile and rapid methodology for monitoring the oxidation of edible oils as a function of the oil’s chemical and physical/morphological arrangements [4,5,6,7,8,9]. LF NMR is also under intense development for many other applications, such as pharmaceuticals, material science, geology, and agriculture [10,11,12,13]. We have focused on developing a portable LF ^1^H NMR designated time-domain (TD) NMR sensor for the measurement of ^1^H relaxation times of both the chemical and physical structure/composition of intact food samples. These ^1^H relaxation times can efficiently monitor a food’s susceptibility to oxidation at the different stages of preparation, storage, transportation, and digestion [14,15].

LF ^1^H NMR is based on low-field strength magnets that can readily monitor ^1^H protons’ energy relaxation time both with respect to spin–spin (T_2_) and spin–lattice(T_1_) energy relaxation time mechanisms. An important motivation for TD-NMR sensor development is the possibility of direct sample contact without sample modification and the relative simplicity and low cost of the instrumentation vs. commonly used analytical instruments, such as high-performance size-exclusion chromatography (HPSEC), high-field (HF) ^1^H and C^13^ NMR, FTIR, gas chromatography, and mass spectroscopy (GC-MS) [11,16,17,18,19]. In addition, sample modification is needed for the wet chemical analysis of extracted food samples, such as in the peroxide value (PV), *para*-anisidine value (PAV), and total oxidation (TOTOX) values used to determine the oxidized products, and the samples’ oxidative stability [20,21,22]. An important potential aspect of TD-NMR sensors is their rapid sample measurements and data generation, with applications for direct online analysis of a product during preparation, storage, or cooking. In this regard, LF ^1^H NMR, spin–spin energy relaxation measurements (T_2_), and reconstruction algorithms for data generation are on the order of 1 min and have an advantage over spin–lattice (T_1_) measurements, which are on the order of tens of minutes. In this paper, we highlight the developments of TD-NMR sensor analysis with respect to rapid online analysis of the oxidation of vegetable oils with saturated, monounsaturated, and polyunsaturated alkyl chains [14]. The rapid online monitoring of an oil’s oxidative modifications was achieved using the T_2_ determination of the samples’ average, self-diffusion (D) value, and their changes upon oxidation [23]. The D values of the samples were also rapidly determined for intact samples. This methodology can readily characterize and quantify the different oxidative stages of fatty acid (FA) oils by correlating the D values with the oil’s peroxide, aldehyde, and total oxidation (TOTOX) values [15], and can be monitored in less than a minute in intact non-modified samples as compared to several hours using conventional colorimetric assays [24,25,26,27,28,29].

The thermal oxidation of oils is a mechanism that includes initiation, propagation, and termination steps, generating basic chemical and structural changes during a food’s preparation, storage, and cooking. The process starts with an induction time prior to ^1^H abstraction, which initiates the propagation of toxic peroxides, aldehydes, and polymerization end products [11,19,30,31,32]. Common methodologies for an oil’s analysis are PV; PAV; TOTOX—GC; high-field (HF) ^1^H NMR, but as described above and in previous papers [16,19,20,21,22], TD-NMR sensor application is being developed for a more rapid direct analysis of oils’ oxidation. Both the T_1_ and T_2_ values of TD-NMR can form 1D (T_1_ or T_2_ values) and 2D (T_1_ vs. T_2_) material graphic spectrums. Upon oxidation, the changing chemical and morphological material arrangements are monitored quantitatively by the T_1_ vs. the T_2_ peak shift and bending [4,33,34,35]. As described above, upon oxidation, the T_1_ and T_2_ shifts also correspond to changes in the materials’ self-diffusion (D) and viscosity, which were readily calculated from the same oil samples used to calculate the T_2_ values from the TD-NMR. The D values correspond to oxidative chemical and structural changes, such as aldehyde formation [36,37,38] and polymerization [31]. The aim of the present study was to measure an oil’s D values from TD NMR analysis for a more rapid and accurate determination of the oxidation status of edible oils.

## 2. Materials and Methods

### 2.1. Experimental Design

Thermal oxidation was carried out on samples of butter, coconut oil, olive, canola, soybean, and linseed oils that were purchased from local suppliers. The autoxidation experimental design was based on previous studies [39,40], wherein autoxidation was induced by heating 100 mL of the sample in a 250 mL beaker on an 80 °C hot plate. Air was pumped into the beaker with maximum stirring for 120 h, using a glass Pasteur pipette and a vacuum pump (Vacuubrand MZ 2C Diaphragm Vacuum Pump, Merck, Darmstadt, Germany). Initially and after 24, 48, 72, 96, and 120 h of the oxidation process, a 10 mL sample was removed for analysis. The analysis included commonly used chemical analysis of the peroxide values (PVs) and aldehyde values (*para*-anisidine) and the calculation of the total oxidation values (TOTOX) [24,25,26,27,28].

### 2.2. Methodologies

The spin–spin ^1^H LF NMR measurements were carried out on a Maran bench-top pulsed NMR analyzer (Resonance Instruments, Witney, UK) with a permanent magnet and an 18  mm probe head operating at 23.4  MHz. Before each measurement, the sample was stabilized at 40 °C for 20  min, and then quickly equilibrated inside the instrument. The spin–spin energy relaxation time constants (T_2_) were generated using a Carr–Purcell–Meiboom–Gill (CPMG) pulse sequence, and the spin–lattice energy relaxation time constants (T_1_) were generated using an inversion recovery pulse sequence.

The signal processing was based on a PDCO inverse Laplace transform (ILT) optimization algorithm with alpha = 0.5 [41,42]. Four replicates of each sample were tested and a mean was generated for each T_2_.

The self-diffusion measurements were carried out with a 20  MHz mini spec bench-top pulsed NMR analyzer (Bruker Analytic GmbH, Karlsruhe, Germany) equipped with a permanent magnet and a 10-mm temperature-controlled probe head, according to [40].

The self-diffusion coefficients, the D values, were determined by the pulsed-field gradient spin echo (PFGSE) method [43]. The pulse sequence was performed with 16 scans, τ of 7.5  ms, and a recycle delay of 6  s. Typical gradient parameters were the Δ of 7.5  ms and the δ of 0.5  ms, with time between the 90° pulse and the first gradient pulse of 1  ms, and the G of 1.6  T/m. Each reported value of the self-diffusion coefficient (D) is the average of ten measurements.

The primary oxidation products were evaluated with peroxide value (PV) tests according to the AOAC Official Method 965.33.12 (Official Methods of Analysis of AOAC International, 17th edn., Gaithersburg, MD, USA). While the *para*-anisidine value (PAV) test was used in the assessment of the secondary oxidation products according to the AOCS Official Method Cd 18–90 (2002) [27,44].

### 2.3. Statistical Treatment

The TD-NMR material spectrum graph experiments were repeated at least four times. The self-diffusion coefficient (D) tests were repeated at least 10 times, and the average values ± standard deviations (SD) are given. The conventional colorimetric tests were carried out on all the samples shown in Appendix A and given as average values.

## 3. Results and Discussion

In agreement with previous reports [8,14,15,19,33,34,35], fresh non-heated linseed oil clearly shows the relaxation time/signature of its four major molecular segments assigned to fatty acid’s glycerol, double bonds, aliphatic chains, and alkyl tails in a 2D T_1_-T_2_ graph, reflecting both the oil’s chemical composition and structural self-assembly (Figure 1A). Following 120 h of heating at 80 °C with continuous air pumping, a clear pattern of energy relaxation time shifts along the diagonal line of the T_1_ vs. the T_2_ to lower values, in comparison to non-heated linseed oil, was observed together with a bending effect, in which new peaks appeared with a lower T_2_ and a constant T_1_ (Figure 1B). This last pattern is typical for the termination phase of the oxidation process and is explained as a polymerization end product characterized by significant changes in both the chemical composition and structural arrangement that clearly correlate with an increased viscosity [8,14,45].

The same linseed oil samples used for analysis by the TD NMR sensor (shown in Figure 1) and other edible oils representing saturated (butter and coconut), mono-unsaturated (olive and canola), and poly-unsaturated (soy) oils were analyzed by conventional colorimetric methods of an oil’s oxidation. These methods include the peroxide value (PV), *para*-anisidine value (PAV), and TOTOX. As expected in agreement with previous reports [11,14], the results of Table 1A clearly show that for the saturated butter and coconut oils, minimal changes were obtained in all three tests of thermal oxidation for 120 h.

For the mono-unsaturated olive oil and canola oil, patterns of increased PV, PAV, and TOTOX values were observed. For example, the TOTOX value increased significantly from 21.86 to 124.63 and from 15.48 to 139.37 after 48 h for the olive and canola oils, respectively. These values further increased for these mono-unsaturated oils after 72 and 96 h and peaked after 120 h at 344.58 and 661.69, respectively. It should be noted that the aldehyde levels increased more rapidly in the canola oil samples in comparison to the olive oil. This may be rationalized by the olive oil being produced in a cold-press system, achieving a higher antioxidant level than that in the canola oil produced by solvent extraction [3,19].

As expected, according to well-established oil thermal oxidation theory [2,14,21,32], the poly-unsaturated soy oil and linseed oil had a much faster rate of oxidation in all three oxidation tests in comparison to the previous two types of monounsaturated oils. Following an induction phase of about 24 h, common markers of oxidation, with an emphasis on peroxides (PV), increased after 48–72 h. In the later stages, the aldehyde levels measured by PAV further increased. It should be noted that for the soy oil, the aldehyde level continued to increase until 120 h (295.16), but for the linseed oil, the aldehyde level peaked at 96 h (186.4) and decreased after 120 h (104.9). This late-stage reduction of the aldehyde level in the linseed oil is due to the polymerization process that occurs at this stage of the oxidation process [31,33].

Percent changes in the PV, PAV, and TOTOX from the beginning to the end of the oxidation experiment are shown in Table 1B. For the poly-unsaturated oils, it is clear that the changes and increases in the aldehyde levels dominated the thermal oxidation process at almost all testing times. In the case of the soy oil, after 24 h of heating, the aldehyde level increased by 1520.8% and continuously increased to 45,409.2% after 120 h. The peroxide level increased from 572.4% after 24 h to 3537.8% after 120 h, which is much lower than the aldehyde’s increase. For the linseed oil, due to a longer induction phase, the aldehyde level dominated the oxidation process from 48 h (7764.3%) to 120 h. (6681.5%). This difference between the soy and linseed oils is explained by linseed oil’s polymerization at the end of the process [33]. A similar pattern of change was also observed for the mono-unsaturated canola oil but at a much lower concentration. The aldehyde levels increased by 623.0% after 24 h., and further increased to 20,890.8% after 120 h. In the case of the cold-pressed mono-unsaturated olive oil, an increase in the aldehyde level was observed, but it was lower than the increase in the peroxide level, probably due to the higher content of antioxidants (vitamin E, polyphenols, and phytosterols) (19). A similar pattern of aldehyde product domination of the oxidation process was also observed but at significantly lower rates for the saturated butter and coconut oils. In these cases, the unsaturated fatty acids (18:1, 18:2, 18:3) within the butter and coconut were oxidized, and the levels of aldehydes increased from 195.5% and 143.3 after 48 h to 432.1% and 292.2 after 120 h, respectively.

In agreement with previous reports [33,34,36,38], these data show rapid increases in the aldehyde formation of unsaturated oils during the thermal oxidation process, demonstrating both the changes in the chemical composition as well as the changes in the chemical structural organization. These changes are highly important in the present study and are further supported by the NMR methodologies as discussed below. Aldehyde formation is also correlated with another common and most accurate method for oxidation analysis based on chemical shifts within high-field NMR (^1^H HF NMR) spectrums [36,38,46].

The chemical shift spectra of high-field, ^1^H HF NMR of the same linseed oil samples used in this study show that the glycerol backbone CH_2_ appears between 3.90 and 4.30 ppm, and the glycerol backbone CH appears at 5.17 ppm. Signals from the protons on saturated chains (aliphatic chains) appear between 1.10 and 1.60 ppm, whereas “allylic protons” appear between 1.80 and 2.30 ppm, and “bis-allylic protons” peaks between 2.60 and 2.80 ppm. Olefinic proton peaks appear at 5.20–5.40 ppm and the terminal CH_3_ peaks between 0.70 and 0.90 ppm. This ^1^H HF NMR chemical shift methodology was used to analyze PUFA-rich linseed oil oxidation process by quantifying the reductions in the peak intensities of the unsaturated PUFA and olefin protons in the spectra at 24, 48, 72, 96, and 120 h under thermally induced oxidation at 80 °C plus air pumping. PUFA-rich materials’ oxidation generally occurs via complex free-radical chain reactions characterized by an initiation step, a propagation sequence, and termination steps [15,29,37]. In the initiation step, triggered by the high temperature of 80 °C that the linseed oil was exposed to, a hydrogen radical (H) is abstracted from a PUFA molecule forming alkyl radicals (R) on an allylic carbon, resulting in the isomerization of the double bonds into a conjugated structure [14]. The energy required to remove the hydrogen atom on the alkyl chain is dependent on its position in the molecule, and the hydrogen atom attached to bis-allylic carbon requires the least amount of energy to be removed [29,47]. In the propagation step, the highly reactive PUFA alkyl radicals (R) react with atmospheric oxygen and form peroxy radicals (ROO), propagating a chain reaction. The peroxy radicals (ROO) may react with a hydrogen atom abstracted from another PUFA molecule forming hydroperoxides (ROOH) and another PUFA alkyl radical (R). After 24 h of oxidation, it is possible to see the initial signs of oxidation in the linseed oil ^1^H HF NMR spectrum, with a reduction in the intensity of the peaks of the allylic protons at 2.06 ppm, bis-allylic protons at 2.76 ppm, and olefinic protons at 5.38 ppm. This indicates that the unsaturation of the PUFA molecules of the linseed oil had already started decreasing upon initiation of the oxidation reaction. The decreases in the peaks’ intensity for allylic, bis-allylic, and olefin protons continued after 48, 72, 96, and 120 h, which is rationalized by the development of an oxidation chain reaction.

Furthermore, as described for the linseed oil heated to 70 °C together with air pumping, -OOH hydroperoxide group derivatives were seen at 8.3–8.8 ppm (for cis and trans-conjugated double-bond groups) and at 5.70 and 6.20 ppm (for trans and trans-conjugated double-bond groups) [38].

Aldehyde group derivatives (-CHO) of trans-2-alkenals were at 9.480 and 9.506 ppm, trans and trans-2,4-alkadienals at 9.507 and 9.533 ppm, n-alkanals at 9.748 ppm, 4-hydroxy-trans-2-alkenals at 9.560 and 9.586 ppm, 4-hydroperoxy-trans-2-alkenals at 9.568 and 9.594 ppm, and 4,5-epoxy-trans-2-alkenals at 9.538 and 9.564 ppm [38].

These chemical shifts of thermally oxidized linseed oil samples in ^1^H HF NMR significantly strengthen and support the above-cited evidence and discussion regarding the chemical and morphological changes occurring during thermal oil oxidation processes characterized by LF ^1^H NMR. The generation of the chemical shift peaks and their values in the HF NMR-analyzed samples, however, requires a relatively longer time and significantly higher equipment cost, indicating that this is not an optimal industrial on/at-line analytical procedure for food product manufacturing. A more rapid, potentially on/at-line procedure development for analyzing the degree of oxidation of oil products needed for oil food products’ quality is described below, using the TD-NMR sensor application for fast determination of the self-diffusion values.

TD-NMR determination of self-diffusion (D) is well accepted for characterizing the chemical and physical status of foods with fatty acids and esters [15,23]. The self-diffusion (D) values of a list of edible oils thermally oxidized at 80 °C with air pumping for 120 h are shown in Table 2. As expected from the results shown above, Table 1A and Table 2 clearly show that in the saturated butter and coconut oils, there was almost no oxidation, and very small changes in the D values were measured, suggesting that almost no chemical or morphological changes were detectable. In the mono-unsaturated oils, a relatively small decrease in the D level was seen, starting with 0.030 for both the fresh olive and canola oils and ending after 120 h at 80 °C, with 0.026 and 0.019 for the olive and canola oils, respectively. The relatively smaller reduction in the D values in the olive oil is rationalized by the higher content of antioxidants than in the canola oil. This correlates with the data shown in Table 1A. These results suggest a correlation between the relatively slow oxidation rates of these two mono-unsaturated oils, as shown in Table 1A, and their self-diffusing rate. However, the poly-unsaturated soy and linseed oils clearly show a dramatic decrease in their D values from 0.034 to 0.012 and from 0.040 to 0.018, respectively, during the 120 h of thermal oxidation. The fact that the starting and final D levels of the soy oil were higher than for the linseed oil is explained by the difference in the freshness and method of extraction of these two oils (soy oil by solvent extraction and linseed oil by cold extrusion) [8].

The data shown in Figure 2 demonstrate a correlation between the self-diffusion values (D) of the linseed oil and the three parameters analyzed by the common conventional tests of PV, PAV, and TOTOX using the same linseed oil samples during the same period of thermal oxidation. It should be noted that in agreement with previous reports [14,33], at the last testing point of 120 h, the linseed oil tended to polymerize and formed a viscous gel-like structure that significantly decreased the D values. Therefore, the data in Figure 2 are limited to 96 h. The linear fitness line (R^2^) for PV vs. D is 0.8415, PAV vs. D is 0.8636, and TOTOX vs. D is 0.849. Furthermore, the correlation rate was even somewhat higher at the testing point of 96 h when the level of polymerization was lower. The best correlation was found for PAV (aldehydes) vs. D, suggesting a better relationship correlating the proton mobility/movement with the D values within the linseed oil and aldehyde formation, which represents the oils’ chemical–structural changes during the initial stages of oxidation [33,36].

The suitability of the self-diffusion D values for monitoring the oxidation status of edible oils, other than linseed oil, was confirmed when comparing the PAV to the D (not shown) of these various edible oils during the same times of thermal oxidation at 80 °C. A good correlation was obtained for the self-diffusion of the highly oxidized oils, including soy, linseed, and canola, with R^2^ values of 0.9356, 0.8636, and 0.8813, respectively. As expected, lower aldehyde/D ratios were obtained with the less oxidized (Table 1A) olive oil, coconut oil, and butter. Regarding the last two saturated oils, which are known to be less sensitive to thermal oxidation, these data are easy to rationalize. The relatively low aldehyde/D ratio for the mono-unsaturated olive oil is more difficult to understand, but according to the previous results shown above, it seems that it is due to the relatively low level of aldehyde formation in the period of the experimental design used in the present study.

Based on the good correlation between self-diffusion (D) and aldehyde *para*-anisidine values (PAV), the linseed oil samples of the present thermal oxidation experiment appeared to form three groups, divided into green, yellow, and red, of oxidized molecular categories (Figure 3). The green group included non-oxidized safe oils with a D range of 0.036–0.047 × 10^−9^ m^2^/s and a PAV range of 0–40 mmol/kg; the yellow group included partially oxidized medium oils with a D range of 0.026–0.035 × 10^−9^ m^2^/s and PAV range of 40–150 mmol/kg; and the red group included highly oxidized unsafe oils with a D range of 0.012–0.025 × 10^−9^ m^2^/s and a PAV range of 90–220 mmol/kg. It should be noted that there were some outlier values that seem to have been due to experimental and/or analytical errors.

We used linear models (Table 1 and Figure 1) to compare different oxidative treatments (measured in terms of hours of oxidation) and the corresponding conventional laboratory tests such as the *para*-anisidine, peroxide, and total oxidation values vs. the self-diffusion coefficients as determined by a TD-NMR sensor. The linear approximations were sufficiently accurate, so the polynomial regressions were not discussed. Table 1 shows the expected change in each measurement per unit increment (i.e., 1 h) of oxidative treatment. For example, the diffusion coefficients are expected to show a decrement of −1.882 × 10^−4^ (95% CI −2.069 × 10^−4^–−1.695 × 10^−4^) for each additional hour of oxidative treatment. The R^2^/R^2^ adjusted values show that all the models could capture at least 66% of the variance in the data. Considering that the oxidative treatment of vegetable oils and the corresponding laboratory measurements are both associated with some inexplicable variance, the results are highly homogeneous in such a way that all four different lab methods clearly converge to the same accuracy level. In particular, the diffusion coefficients reflect the oxidation treatment with an accuracy equivalent to the conventional methods (*para*-anisidine, peroxide, and total oxidation). An important characteristic of edible oils containing mono- and poly-unsaturated fatty acids is the extent of oxidized products within the foods, as these are often toxic materials. Oils are readily oxidized during the stages of preparation, storage, and cooking, for which industrial rapid and accurate on/at-line analysis of the extent of oxidation is not available by the current conventional methods, such as PV, PAV, etc.

## 4. Conclusions

In this paper, we demonstrate a rapid on/at-line industrial TD-NMR sensor application based on rapid and accurate self-diffusion (D) value determination for the evaluation of edible oil’s oxidation status without sample modifications. The D values correlate well with the aldehyde concentrations, which is in line with the chemical and morphological changes of the oil samples and is an excellent marker of the samples’ oxidation statuses. These results indicate that conventional methodologies used for determining an oil’s oxidation status, such as PV for peroxides and PAV for aldehydes, can be substituted with a much more rapid determination of the food’s oxidative status based on TD-NMR sensor determination of the average self-diffusion D values. This is demonstrated for edible oils, including saturated, mono-unsaturated, and poly-unsaturated oils. The results clearly show that saturated butter and coconut oil had minimal changes within all stages of thermal oxidation. For mono-unsaturated olive oil and canola oil, there was a pattern of significantly increased oxidation after 48 h. Poly-unsaturated soy oil and linseed oil had a much faster rate of oxidation. Linseed oil’s oxidation status indicator/marker can be formulated from the data as follows: (a) non-oxidized—GOOD, with a D > 0.035; (b) partially oxidized—MID, with a D range of 0.035–0.025; and (c) complete oxidation—highly oxidized with a D < 0.025. This D signature approach can be used for pattern recognition (PR) and profiling of the oxidation status and opens the way for machine learning (ML) and developing fast and accurate semi-autonomic TD NMR sensor applications for food product oxidation at the industrial scale.

## 5. Patent

Wiesman, Z., Campisi-Pinto, S., Osheter, T., Linder, C., Osheter, A., Semi-Autonomic TD NMR Sensor of Food Safety and Quality, US provisional patent Application No. 11-246 registered by BGN 23 March 2022.

## Figures and Tables

**Figure 1 molecules-27-06064-f001:**
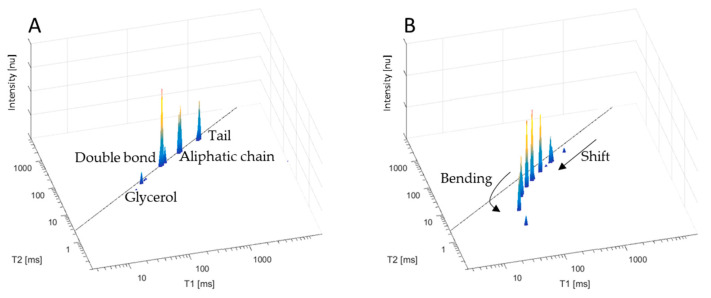
Chemical and morphological TD NMR sensor 2D T_1_-T_2_ relaxation times of linseed oil before (**A**) and after 120 h of thermal oxidation at 80 °C plus air pumping (**B**).

**Figure 2 molecules-27-06064-f002:**
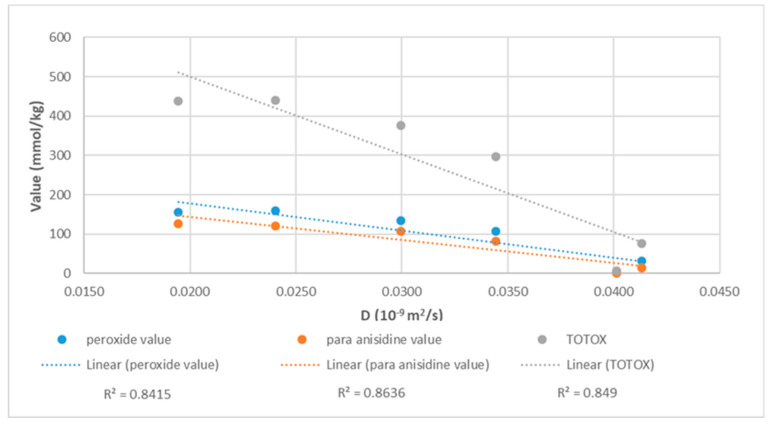
Correlation between linseed oil self-diffusion (D) and PV/AV/TOTOX. Averages of 38 analyses for each point taken from the database (S1) were used.

**Figure 3 molecules-27-06064-f003:**
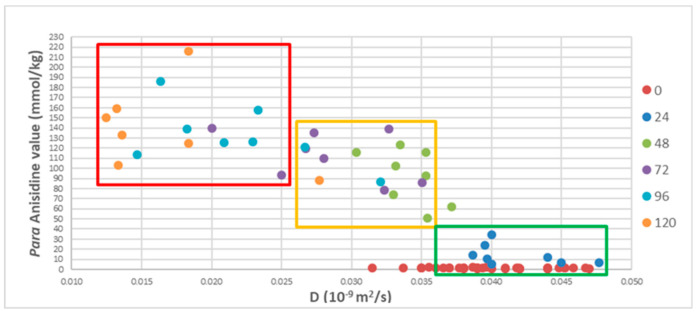
Relationship between *para*-anisidine values (PAV) and diffusion coefficient (D) of linseed oil during thermal oxidation at 80 °C. Averages of 38 analyses for each point taken from the database (S1) were used.

**Table 1 molecules-27-06064-t001:** Conventional analyses of rates of oxidation of saturated, mono-unsaturated, and poly-unsaturated edible oils during thermal oxidation at 80 °C plus air pumping. (A) PV, PAV, and TOTOX average values of edible oils during thermal oxidation. Linseed oil is presented as average ± SD. More details of all the oils are available in the Appendix A, Appendix A. (B) Changes in percentages of PV, PAV, and TOTOX from the beginning of the experiment.

**A**	**Time of Heating** (**h**)	**0**			**24**			**48**			**72**			**96**			**120**		
		**PV**	**PAV**	**TOTOX**	**PV**	**PAV**	**TOTOX**	**PV**	**PAV**	**TOTOX**	**PV**	**PAV**	**TOTOX**	**PV**	**PAV**	**TOTOX**	**PV**	**PAV**	**TOTOX**
		**(mmol/kg)**			(**mmol/kg)**			**(mmol/kg)**			**(mmol/kg)**			**(mmol/kg)**			**(mmol/kg)**		
	**Saturated FA:**																		
	Butter	6.9	1.12	14.93	9.14	1.39	19.68	11.53	2.19	25.26	8.53	2.33	19.4	7.32	3.12	17.75	8.66	4.84	22.17
	Coconut oil	8.18	0.9	17.25	7.03	1.2	15.26	7.83	1.29	16.96	6.54	1.38	14.45	5.46	2.05	12.96	6.11	2.63	14.84
	Mono:																		
	Olive oil	6.48	8.9	21.86	16.44	10.27	43.16	56.08	12.46	124.63	107.38	12.9	227.66	62.12	12.77	137.02	155.24	34.09	344.58
	Canola oil	7.31	0.87	15.48	16.44	5.42	38.31	56.44	26.48	139.37	107.38	41.18	255.94	206.53	157.45	570.5	239.97	181.75	661.69
	**Poly:**																		
	Soy oil	7.38	0.65	15.41	40.6	9.82	91.01	123.72	31.94	279.38	237.05	126.58	600.68	237.62	233.56	708.81	252.86	295.16	800.87
	Linseed oil	3.3± 0.41	1.57 ± 0.26	8.19 ± 0.83	33.8± 4.46	6.72 ± 1.09	74.3 ± 8.92	192.8± 12.58	121.9 ± 25.74	501.2± 25.17	187.5 ± 32.59	136.5 ± 35.83	514.8 ± 65.18	196.4 ± 15.95	186.4 ± 52.26	579.1 ± 31.9	129.8 ± 9.21	104.9 ± 22.67	362.3 ± 18.43
**B**	**Time of heating (h)**	**0**			**24**			**48**			**72**			**96**			**120**		
		PV	PAV	TOTOX	PV	PAV	TOTOX	PV	PAV	TOTOX	PV	PAV	TOTOX	PV	PAV	TOTOX	PV	PAV	TOTOX
	**Saturated FA:**																		
	Butter	100	100	100	132.5	124.1	131.8	167.1	**195.5**	169.2	123.6	**208.0**	129.9	106.1	**278.6**	118.9	125.5	**432.1**	148.5
	Coconut oil	100	100	100	85.9	**133.3**	88.5	95.7	**143.3**	98.3	80.0	**153.3**	83.8	66.7	**227.8**	75.1	74.7	**292.2**	86.0
	**Mono:**																		
	Olive oil	100	100	100	253.7	115.4	197.4	865.4	140.0	570.1	1657.1	144.9	1041.4	958.6	143.5	626.8	2395.7	383.0	1576.3
	Canola oil	100	100	100	224.9	**623.0**	247.5	772.1	**3043.7**	900.3	1468.9	**4733.3**	1653.4	2825.3	**18**,**097.7**	3685.4	3282.8	**20**,**890.8**	4274.5
	**Poly:**																		
	Soy oil	100	100	100	572.4	**1510.8**	614.3	1566.7	**14**,**144.6**	1716.0	3677.6	**19**,**473.8**	4382.3	3337.4	**35**,**932.3**	4791.5	3537.8	**45**,**409.2**	5405.8
	Linseed oil	100	100	100	1024.2	428.0	907.2	5842.4	**7764.3**	6119.7	5681.8	**8694.3**	6285.7	5951.5	**11**,**872.6**	7070.8	3933.3	**6681.5**	4423.7

**Table 2 molecules-27-06064-t002:** Self-diffusion (D) values of saturated, monounsaturated, and polyunsaturated edible oils during thermal oxidation stimulation conditions at 80 °C (average + SD is presented).

Time of Heating (h)	0	24	48	72	96	120
**Saturated**	**D (10^−9^ m^2^/s)**					
Butter	^1^ 0.161 ± 0.0100	0.026 ± 0.0016	0.033 ± 0.0012	0.029 ± 0.0022	0.033 ± 0.0022	0.029 ± 0.0008
Coconut Oil	0.037 ± 0.0014	0.038 ± 0.0033	0.036 ± 0.0005	0.036 ± 0.0016	0.035 ± 0.0039	0.037 ± 0.0029
**Mono-Unsaturated**	**D (10^−9^ m^2^/s)**					
Olive Oil	0.030 ± 0.0009	0.028 ± 0.0024	0.025 ± 0.0031	0.024 ± 0.0025	0.024 ± 0.0014	0.026 ± 0.0012
Canola Oil	0.030 ± 0.0033	0.027 ± 0.0005	0.028 ± 0.0022	0.023 ± 0.0012	0.019 ± 0.0014	0.019 ± 0.0033
**Poly-Unsaturated**	**D (10^−9^ m^2^/s)**					
Soy Oil	0.034 ± 0.0021	0.029 ± 0.0034	0.028 ± 0.0033	0.022 ± 0.0023	0.018 ± 0.0025	0.012 ± 0.0008
Linseed Oil	0.040 ± 0.0038	0.041 ± 0.0037	0.034 ± 0.0040	0.030 ± 0.0054	0.024 ± 0.0070	0.018 ± 0.0069

^1^ Solid at room temperature.

## Data Availability

Not applicable.

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
