# Peer review of "1H LF-NMR Self-Diffusion Measurements for Rapid Monitoring of an Edible Oil’s Food Quality with Respect to Its Oxidation Status"

_molecules, 2022, doi:10.3390/molecules27186064_

Round 1

Reviewer 1 Report

The manuscript is aimed to investigate new reliable and rapid methodologies for the determination of the oxidation status of edible oils based on the measures of proton 1H spin-spin (T2) and spin-lattice(T1) energy relaxations by bench-top low field pulsed NMR spectrometers.

The rational approach is sound and well presented, as well as the experimental procedures that appear to be appropriate to the scope of the manuscript. Extensive set of experimental data are clearly reported, and conclusions appear to be consistent and open the way to a practical application of the investigated methodologies.

I have just a few short notes.

1) Page 7, lines 208-210: I found inappropriate (unclear) the use of “unsaturated protons” and of “poly-unsaturated fatty acids (PUFA) peaks”. I suggest using “allylic protons” and “bis-allylic protons” as done in page 8, line 228.

2) Page 9, line 266: D values …………… “0.04” should be “0.040”

 Some minor English editing might be required

Author Response

Please find below our detailed response to each comment of reviewer 1:

Reviewer 1:

1) Page 7, lines 208-210: I found inappropriate (unclear) the use of “unsaturated protons” and of “poly-unsaturated fatty acids (PUFA) peaks”. I suggest using “allylic protons” and “bis-allylic protons” as done in page 8, line 228.

Response – Indeed we accept the comment of the reviewer and corrected the text accordingly.

2) Page 9, line 266: D values …………… “0.04” should be “0.040”

Response - Indeed we accepted the comment of the reviewer and corrected the text accordingly.

3) Some minor English editing might be required

Response – As was suggested by the reviewer, we carefully went over the English all over the manuscript and significantly improved it.

Reviewer 2 Report

Dear authors, 

I have completed my review of the submission entilted "1H LF-NMR Self-Diffusion Measurements for Rapid Monitoring of an Edible Oil’s Food Quality with respect to its Oxidation Status". This is an important submission, in which, the authors proposed an analytical method to monitore edible oil oxidation status based on TD-NMR sensor. I see that the manuscript is worth to be considered at Molecules. However, some major revisions have to be incorporated to improve the manuscript. Please find below, my suggestions and comments: 

1- A careful revision must be done to correct typo and language mistakes, 

2- The state-of-art of research has to be deepened in light of published on edible oil oxidation. Here is an important review article to be added and discussed: https://doi.org/10.3390/su14020849,

3- Line 80, the authors must highlight gaps of knowledge before highlgihting the objectives of the work, 

4- Abbreviations have to be defined in full at first mention. Please revised accordingly throughout the submission, 

5- In Fig 1, the intensity unit must be defined, 

6-  Line 173, please use past tense and revise the entire Results section acordingly,

7- To present your outcomes, please add Standard Deviation and stastical significance differences/similarities,

8- Lines 255-257, please check these sentences,

9- In Fig. 2,  please check typo mistakes,

10- In the same figure, please correct the imprecisions regarding R2, it is expressed as %.

Recommendation: Accept after major revisions.

Kind regards.

Author Response

Please find below our detailed response to each comment of reviewer 2:

Reviewer 2:

1- A careful revision must be done to correct typo and language mistakes

Response – Indeed we accepted the comment and suggestion of the reviewer and significantly revised the manuscript with emphasis and special care of the typo and language mistakes corrections all over the manuscript.

2- The state-of-art of research has to be deepened in light of published on edible oil oxidation. Here is an important review article to be added and discussed: https://doi.org/10.3390/su14020849

Response Indeed we accepted the comment and suggestion of the reviewer and this important and relevant new review article in several places in the text and added it to the list of references.

"Fadda, A.; Sanna, D.; Sakar, E.H.; Gharby, S.; Mulas, M.; Medda, S.; Yesilcubuk, N.S.; Karaca, A.C.; Gozukirmizi, C.K.; Lucarini, M.; et al. Innovative and Sustainable Technologies to Enhance the Oxidative Stability of Vegetable Oils. Sustainability 2022, 14, 849. https://doi.org/10.3390/su14020849"

3- Line 80, the authors must highlight gaps of knowledge before highlgihting the objectives of the work

Response – As was suggested by the reviewer we emphasized before the objectives of the present study that there is a need to develop a more rapid direct analysis of oil’s oxidation. This requirement is not well addressed by the available methodologies and common technologies listed in the text.

4- Abbreviations have to be defined in full at first mention. Please revised accordingly throughout the submission

Response – We accepted this comment of the reviewer and defined the abbreviations in full at first mention throughout the submission.

5- In Fig 1, the intensity unit must be defined

Response – We accepted the comment of the reviewer and added the intensity unit in Fig 1.

6- Line 173, please use past tense and revise the entire Results section acordingly

Response – We accepted the comment of the reviewer and accordingly corrected the text in the revised version of the manuscript.

7- To present your outcomes, please add Standard Deviation and stastical significance differences/similarities

Response – As suggested by the reviewer we added Standard Deviation in Table 1A for main used Linseed oil. In order to keep the ease of reading of the results of this Table we did not add SD for other oils. In any case we added to the title of this Table a comment that additional details of all the oils can be found in the Supplemental material attached to the article.

8- Lines 255-257, please check these sentences

Response – As suggested by the reviewer we checked and revised these sentences in the revised submission.

9- In Fig. 2,  please check typo mistakes

Response – As suggested by the reviewer we checked and revised the typo mistakes in the revised submission.

10- In the same figure, please correct the imprecisions regarding R2, it is expressed as %.

Response – As suggested by the reviewer we checked again the way R2 is presented in Fig 2 and got to a conclusion not to present it in % and to leave it in the original way that is most common in the scientific literature.

Round 2

Reviewer 2 Report

Dear authors, 

Thank you for taking my comments and suggestions into account. It seems that the manuscript has been improved and therefore I recommend its publication in Molecules. 

Kind regards.